# Recent Advances in Functional Polyurethane and Its Application in Leather Manufacture: A Review

**DOI:** 10.3390/polym12091996

**Published:** 2020-09-02

**Authors:** Saiqi Tian

**Affiliations:** College of Education, Wenzhou University, Wenzhou 325035, China; tiansaiqi@wzu.edu.cn; Tel.: +86-0577-8668-9665

**Keywords:** functional polyurethane, leather, recent advances, review

## Abstract

Over last few years, polyurethane (PU) has been applied in a number of areas because of its remarkable features, such as excellent mechanical strength, good abrasion resistance, toughness, low temperature flexibility, etc. More specifically, PU can be easily “tailor made” to meet specific demands. This structure–property relationship endows great potential for use in wider applications. With the improvement of living standards, ordinary polyurethane products cannot meet people’s growing needs for comfort, quality, and novelty. This has recently drawn enormous commercial and academic attention to the development of functional polyurethane. Among the major applications, PU is one of the prominent retanning agents and coating materials in leather manufacturing. This review gives a summary of academic study in the field of functional PU as well as its recent application in leather manufacture.

## 1. Introduction

Polyurethane (PU) was first introduced by a German professor (Professor Dr. Otto Bayer) and his co-workers in the 1940s [1], and has been applied in a very broad range of commercial and industrial fields due to its unique combination of unusual features including excellent mechanical strength, good abrasion resistance, toughness, low temperature flexibility, corrosion resistance, processability, etc. The basic repetitive unit in PUs is the urethane group (–NHCOO–), which is produced from the reaction between isocyanate (–NCO), polyols (–OH), and other additives [2]. Segmented polyurethanes are composed by two blocks: the soft segment forms by a macrodiol (polyether or polyester diol), and the hard segment is composed by a diisocyanate and a low molecular weight chain extender or crosslinkers [3].

Generally, PU’s structure is determined by the attributes of the raw materials, such as whether they contain hard or soft segments, their molecular weight, polydispersity, and crosslinking ability. It can be easily designed by changing the types and quantities of isocyanate, polyol, surfactants, catalysts, fillers, and matrices during the manufacture process or via advanced characterization techniques, so as to meet the desired performances [4,5]. Therefore, polyurethane products show various types, including elastomers, sheets, adhesives, coatings, and foams.

With the improvement of living standards, ordinary polyurethane products cannot meet people’s growing need of comfort, quality, and novelty. Consequently, significant interest has been directed to the development of functional polyurethane [6,7]. Normally, functional polyurethanes can show stimuli response to the environment or possess unique characteristics, like thermosensitivity, shape memory, self-healing, and photochromicity.

Overall, there are two widely used approaches to obtain functional features: One is to adjust the structure of PU by controlling raw materials, hard and soft segment, molecular weight, polydispersity, and crosslinking ability. The other one is to incorporate of functional molecules into PU backbones or networks. Accordingly, PU can not only maintain its own properties but also acquire the characteristic of functional groups.

Among the major applications, PU plays an important role during leather manufacturing, which is mainly used in retanning and finishing process Waterborne PU is one of the best candidates for leather retanning. Owing to the similarity between the urethane group in PU and the peptide chain in collagen, the resultant leathers retanned by PU can keep the feeling of genuine leather [8]. PU is also desirable coating materials in finishing, exhibiting many preeminent properties, such as excellent flexibility, superior handling, and adhesive strength [9]. It is found to be capable to hide crust leather defects or irregular appearance and confer specific properties. Nowadays, leather products are required to have higher comfort, quality, and novelty, which can be realized by functional PU.

To the best of our knowledge, no effort has been made in the literature compiling functional PU and its application in leather manufacturing. The main objective of this review is to give a summary of the academic study in the field of functional PU as well as its recent application in leather manufacture. This is illustrated by reviewing preparation methods for the realization of polymeric materials and mechanism of these unique performances. This review will report a substantial number of examples that we considered suitable to provide readers with illuminating information about the main topic. The examples here reported come predominantly from recent relevant literature.

## 2. Functional Polyurethane

Varieties of functional polyurethane have been made into commercial production. However, most still need to be explored and improved. According to the application, the five important categories PUs are (1) anti-fouling, (2) self-healing, (3) anti-bacterial, (4) luminescent and color-tunable, and (5) shape memory.

### 2.1. Anti-Fouling PU

Inspired by the super-hydrophobic surface of the natural lotus leaf, the anti-fouling PU can be realized through two approaches: decreasing surface free energy and increasing surface roughness.

Surface free energy of materials is determined by the chemical composition. As a rule, PU is modified by organosilicone or organofluorine.

Attributed to its unique structure, organosilicone has the features of both organics and inorganics. The non-polarity of –Si–O– in its backbone results in fairly low surface tension, ensuring remarkable hydrophobicity. Usually, polysiloxane is covalently incorporated into PU chains to obtain a block copolymer. For example, Rahman et al. [10] prepared polysiloxane–polyurethane through a prepolymer process and investigated the potential application of it in marine coatings. Polydimethylsiloxane (PDMS) and poly (tetramethyleneoxide glycol) were used as the soft segments. Results showed that the coating was smooth on account of the surface enriched by silicone in PDMS with a content of 15.76 wt.%. This effectively prevented marine fouling in sea water. Additionally, siloxane–polyurethane was prepared and the effect of the attachment of silicone oils into the fouling–release coating system was explored by Galhenage and coworkers [11]. A thicker interfacial layer was formed by phenylmethyl silicone oil, resulting in improved hydrophobic performance.

In recent years, there has been an increasing interest in fluorinated polymers with unique low surface energy, high hydrophobicity, and non-sticking behavior. Generally, to introduce organofluorine units into polyurethane, the fluorine unit was embedded into the backbone of polyurethane chain through a covalent bon [12,13,14]. Wen [15] prepared a fluoro alcohol-terminated isocyanate trimer by 3,3,4,4,5,5,6,6,7,7,8,8,8-tridecafluoro-1-octanol and hexamethylene diisocyanate trimer, and then fabricated a series of fluorine-containing water-based polyurethanes (Figure 1). By controlling the fluorine content, the PU was endowed with low surface free energy and great wetting ability. Next, a different fluoroalcohol-terminated isocyanate trimer was utilized to synthesize a series of fluorinated waterborne polyurethanes with various lengths of fluorinated side chains [16]. It is indicated that the long fluorocarbon chain plays an important role in the molecular design of low surface energy polyurethane coating materials.

According to the Cassie–Baxter equation [17], the hydrophobicity of materials increases with increasing surface roughness. In current research, hydrophobicity of PU was achieved by a hierarchical morphology comprising of packed nanoparticles, which was induced in thermoplastic polyurethane (TPU)/silica nanocomposite coatings (Figure 2) [18]. After a certain processing time, the formation of a greatly packed morphology by silica nanoparticles appeared, which lead to a significant surface roughness. Through the roughness studies, the sample processed at a pressing time of 5 min was proved to display higher roughness parameters than the one processed at 1 min. The presented pressing method has promising potentials even in turning hydrophilic polymers like TPU into superhydrophobic surfaces with self-cleaning behavior.

Particularly, incorporation of functional units into PU matrices can realize self-cleaning behavior. For instance, nano-TiO_2_ is a highly efficient photocatalyst to decompose dirt, on account of its low cost, non-toxicity, and high stability [19,20,21,22]. In the application of self-cleaning polymer coatings, the introduction of photocatalytic activity can endow self-cleaning ability [23]. In a research, it provided a novel and eco-friendly approach for fabrication of waterborne polyurethane acrylate with self-cleaning performance in photocatalysis by incorporating surfactant-modified TiO_2_/reduced grapheneoxide (TiO_2_/rGO) nanocomposites [24]. The dispersibility of TiO_2_/rGO nanocomposites in the polymer matrix was improved by incorporation of cationic surfactant hexadecyl trimethyl ammonium bromide (CTAB). Methyl orange (MO; 88.3%) was selected as the model dirt. After 6 h of illumination by visible light, MO was decomposed in the existence of the sample of 0.5% C-TiO_2_/rGO-WPUA. This WPUA composites presented appealing self-cleaning ability in photocatalysis.

### 2.2. Self-Healing PU

Self-healing materials are able to recover their fundamental properties after damage has occurred [25,26,27,28]. This ability in materials increases lifetimes, and is especially important to perform in a designed manner for significant times where repair is not possible. Imparting self-healing functionalities to PU may be achieved through two different approaches: blended and intrinsic methods [29]. For blended methods, healing agents and catalysts are embedded into the PU matrix through capsule or vascular, the cracks of which can induce polymerization of healing agents, realizing the recovery of damage. Conversely, a limited number of healing agents and catalysts cannot repair PU repeatedly. Furthermore, the quantity of healing agents and catalysts in PU is finite, which cannot repair PU repeatedly, after they are consumed. Thereby, intrinsic methods become a desirable approach to prepare self-healing PU. In this case, repairing is achieved through the inherent reversibility of bonding in PU backbones, which acts as a healing agent, ideally without external input [25,30]. Intrinsic self-healing behavior is mainly endowed by two types of bonds: One is reversible noncovalent bonds, such as the hydrogen bond and π–π bond. The other one is covalent bonds, such as the bond formed by Diels–Alder (DA) reaction and disulfide bond.

Hydrogen bond is the most common noncovalent bonds with the feature of selectivity, cooperativity and reversibility. For example, a self-healing covalent PU is successfully prepared, which contains urea, amide, and urethane groups with plenty of hydrogen-bonding sites, offering numerous lateral interactions between the polymeric chains [23]. A good proportion of the hydrogen-bonding network between the polymer chains is extremely stable even at high temperatures, which provides appealing autonomous self-healing ability. A healing efficiency of approximately 85% after heating to 100 °C for 24 h is achieved.

The DA reaction is a thermo-induced [4 + 2] cycloaddition reaction by furan group and maleimide group, which are generally utilized as a typical diene and dienophile, respectively (Figure 3). Cleavage reaction (r-DA reaction) of the DA adducts occurs at high temperature, subsequently regenerating the corresponding diene and dienophile. Once the temperature reduces sufficiently, the DA reaction takes place, forming a DA adduct [31,32]. DA/retro-DA chemistry is regarded to be simple and effective, occurs under mild reaction conditions, and has minimal side reactions to build self-healing systems [33,34,35]. In a research, special attention was given for the a facile route to design and prepare a self-healing and recycling PU based on reversible DA/retro-DA reactions [36]. A study was carried out on covalently conjugation of a novel diol containing DA bonds in a PU backbone. After being cut into two pieces, the PU films were thermally treated at 130 °C for 30 min. It was confirmed that the molecular chains of PU were cleaved into small molecular weight fragments through retro-DA reactions. When exposed to 65 °C for 24 h, most of the dissociative maleimide and furan moieties relinked again via DA reaction. The mechanical performance was restored, which was attributed to hydrogen bonds surrounding the cracked location. PU presented the self-healing efficiencies of 92.5% with the 15.6% solid content of DA diol (Figure 4).

Disulfide bonds can initiate chain exchange reaction under exposure to heat, UV light, and redox conditions [37,38,39,40] (Figure 5). More specifically, disulfide exchange can be activated at moderate temperatures (about 60 °C) and without external stimuli, able to endow disulfide-based PU efficient room temperature self-healing features [41,42,43,44]. Kim [38] proposed a transparent and easily processable polyurethane (IP-SS) using bis (4-hydroxyphenyl) disulfide as the aromatic disulfide component embedded in the hard segments. This PU possesses the highest reported tensile strength and toughness (6.8 MPa and 26.9 MJ m^−3^, respectively). After it is cut in half and reconnected, the mechanical properties are able to recover to more than 75% of those of the original sample within 2 h under room temperature (Figure 6).

Compared to disulfide bonds, diselenide bonds own a lower bond energy (diselenide bonds: 172 kJmol^−1^; disulfide bonds: 240 kJmol^−1^) [46]. This indicates that diselenide bonds can be much more readily induced under milder conditions. Successful synthesis of different PU materials by using di-(1-hydroxyundecyl) diselenide (DiSe) with different ratio as chain extender has been reported [47]. Those materials possessed different healing properties. Under merely visible light, the materials could heal themselves to various extents. Moreover, the healing process could be improved with shorter healing time but superior healing results, with irradiation by directional blue laser. The sample was able to retain its mechanical property and the integrity during the healing process, by using laser without generating heat.

### 2.3. Antibacterial PU

PU can cause the growth of bacteria under certain temperature and humidity conditions during usage and storage [48,49,50]. It can be easily decomposed by bacteria, resulting in age and color changes before breaking. This may essentially threaten human health. Thereby, it is significant to avoid the damage in consequence of microorganism-induced erosion, and prevent the reproduction as well as spread of pathogenic bacteria in PU products.

Antibacterial property can be acquired through introduction of a metal acetate, such as Ag^+^ [51], Cu^2+^ [52], and Zn^2+^ [53]. An example of a Cu–Ag-sputtered PU catheter prepared by Rtimi et al. and an accompanying illustration are presented in Figure 7 [54]. PU catheters with different atomic ratios of sputtered Cu:Ag led to different optical properties and antibacterial activities. The bacterial inactivation dynamics were also investigated, manifesting that the bacterial inactivation time was accelerated to 5 min on a 50%/50% Cu–Ag PU catheter compared to Cu or Ag deposited independently on PU catheters.

In another research, an antimicrobial polyurethane foam was prepared by copper oxide nanoparticles as antimicrobial agents [55]. The PU foams not only maintained good tensile strength, but also were endowed with appealing antimicrobial activity against nosocomial infections. The novel property makes it suitable for applications on antimicrobial hospital mattress to control hospital infections.

Self-antibacterial PU anchors antibacterial groups to its chains chemically, ensuring desirable durability. A study was performed on production of a UV-curable waterborne polyurethane with pendant from 4-NCO pre-polymer and modified by guanidinoacetic acid (GAA). The hydrophilic groups of coating surface were increased by GAA and pendant amine, leading to an optimized antibacterial performance [49]. The novel PU had outstanding properties both in Gram-negative (92.05%) and Gram-positive (94.77%) antibacterial tests. In addition, antibacterial efficiency still maintained 87.94% after 12 times washing.

### 2.4. Luminescent and Color-Tunable PU

There are two types of luminescent PU: fluorescent PU and long afterglow PU. The research of fluorescent PU originates from the exploration of colorful PU. Initially, dyes and pigments are simply mixed with PU matrix. Afterwards, fluorescent PUs were synthesized through chemically embedding fluorescers in PU backbones. Hu’s group [56,57,58,59,60] reported a series of fluorescent PUs by incorporating the different molecular structures of fluorescers into the PU chain, and systematically investigated their fluorescent properties.

Fluorescence is caused by radiation, which ceases almost immediately after the incident radiation stops [60]. Long afterglow is long persistent luminescent phenomenon, whereby the visible luminescent emission remains visible for an appreciable time—from seconds to many hours—after the excitation has stopped [61,62,63]. This facilitates applications of long afterglow PU as night-vision materials in many important fields, such as displays, decorations, traffic signage, medical diagnostics, emergency signs, and military. Nevertheless, long afterglow phosphors are almost always composed of silicate, phosphate, and aluminate activated by rare earth ions. As a result of the weak interaction between inorganic phosphors and organic polymers, physically doping phosphors into PU matrices inevitably lead to incompatibility with the matrix [64,65]. Our groups selected 3-aminopropyltriethoxysilane to encapsulate SrAl_2_O_4_:Eu^2+^,Dy^3+^ phosphors [66]. The surface of the phosphors is decorated by both organic coatings and –NH_2_ groups. Because of the interaction between the –NH_2_ of amino–SrAl_2_O_4_:Eu^2+^,Dy^3+^ and the –NCO of the prepolymer, the compatibility of the two components increases. Thus, the resulting PU obtains better mechanical properties, storage stability, and thermal properties than the phosphors blending sample (Figure 8).

Major long persistent luminescent polymers reported now are blue and green. It makes it difficult to mimic the gorgeous color changes found in nature, therefore limiting its wider applications [67,68]. Thus, we further designed and manufactured a novel PU via incorporating NH_2_–SrAl_2_O_4_:Eu^2+^,Dy^3+^ and the red color conversion agent 1-[(2-hydrocyethly)amino]-anthraquinone [69]. The photoluminescence emission spectrum of phosphors and the excitation spectrum of color conversion agent overlap well. The color conversion agent is proved to be able to be excited by phosphors in the dark. The final PU displays red emission in daylight and emits yellow light in the dark.

Stimuli-responsive materials can be utilized to endow luminescent PU with a more vivid color change. A thermochromic luminescent polyurethane was developed via introducing SrAl_2_O_4_:Eu^2+^,Dy^3+^ phosphors and thermochromic pigment (TP) [64]. TP, which is considered as a smart material, can change its optical properties along with the temperature, based on the the formation or destruction of a colored complex. Under natural light, PU presents as deep red at room temperature and the color disappears when the temperature is higher than 35 °C. In dark, TP can only absorb a small quantity of yellow light, which can be transmitted through polyurethane under 35 °C. The great mass of the green luminescent energy from the phosphors was shielded by the optical barrier formed by TP. After it is warmed to 35 °C, the red optical barrier by TP is removed. As a consequence, the luminescence from phosphors can completely transmit through the polyurethane, and we see the green light with naked eyes (Figure 9).

Mechanochromic materials are able to convert mechanical stimuli into optical signals, and thus the abilities to sense stress and show internal damages are vital to monitor failures like fractures, fatigue, and hysteresis [71,72]. This color change allows adjusting the materials before disastrous failure and further increase in the reliability. An example of thermoplastic polyurethane elastomers (TPU-BBS) with reversible mechanochromic behavior prepared by blended with bis(benzoxazolyl)stilbene dyes is reported in Cellini’s article [67]. Upon large deformation of the polymeric structure, the fluorescence emission shifted from the excimer band to the monomer band, which is caused by reorganization of dye aggregates. The largely reversible mechanochromic behavior is able to controlling by the initial dye concentration in the polymer. Compared with TPU-BBS blends with dye concentrations of 0.1 wt.% and 1.5 wt.%, the one with 0.5 wt.% showed a higher relative variation of the emission ratio during stretching. These results provide a promising reference for preparation of mechanochromic sensors (Figure 10).

### 2.5. Shape Memory PU

Shape memory polymers have the ability to remember their original shape after being deformed and recover it under appropriate stimulus such as temperature, light, electric field, magnetic field, pH, specific ions, or enzyme [74,75]. The incompatibility between soft segments and hard segments in PU leads to microphase separation, which depends on block lengths, hydrogen bonding, and crystallization extent. The hard segments are related to the fixed points or frozen phases, which remain hard during temporary shape. The crystalline melting temperature of the soft segment is the shape recovery temperature. The soft segment is recognized as “molecular switch”, with their crystalline melting temperature being the shape recovery temperature (T_s_). Reversible phase is controlled upon heating above T_s_, and cooling below T_s_, respectively [76,77,78].

The shape-memory behavior of polymers has been known for over half a century and has been widespread in numerous applications. Traditionally, shape memory polymers are triggered by heat. Direct heating is not safe enough and realistic for implementations of many devices, therefore limiting the use more in complex applications. More recently, there is an increasing trend in the exploration of multifunctional shape-memory effects [79,80,81].

The multiple shape effects require at least two segregated domains associated with two distinct thermal transitions for fixing each temporary shape by the corresponding domain. For example, Ban [82] presented a shape memory PU, which was responsive to both UV light and thermal stimuli. 4,4-Azodibenzoic acid (Azoa) was used to build the PUs because Azoa shows typical trans-cis photo-isomerization with light [83]. This novel PU was shape-deformed under UV light and capable of shape fixation in visible light. Finally, the shape recovery was facilitated at higher temperature via weakened hydrogen bond interactions (Figure 11).

pH stimulus is also a good candidate for the design of new shape memory PU. The carboxylic acid in the side chains of waterborne PU is sensitive to pH, which in acid formed dimers, while in alkaline transform from acid to carboxylate to disrupt the dimers. Moreover, the carboxylic dimers are demonstrated to be affected by temperature to dissociate and associate as the temperature rise and down [84]. Based on the above consideration, a polyurethane with thermo-induce triple shape memory effect and pH-sensitive dual shape memory effect was developed [85]. The glass transition of soft segments and the association and disassociation of carboxylic dimers are as two switches to control triple-shape memory property. pH stimulus is realized through the carboxylic dimers to associating in acid and dissociating in alkaline.

## 3. Application in Leather Manufacture

### 3.1. Retanning

Retanning process is honored as the Golden Touch in leather technology, which follows the primary tanning process to overcome some drawbacks of chrome tannage and could help to improve the physical-mechanical properties of leathers. Waterborne polyurethane (WPU) retanning agent, which disperses in aqueous media, is one of the environmentally friendly leather chemicals in leather industry. Carboxyl groups ensure excellent solubility; meanwhile, their coordination with Cr^3+^ immobilizes polymer chains in collagens, so as to disperse fibers well (Figure 12).

Specific properties of final leather can be obtained via retanned by functional WPU. A fluorescent WPU was reported for leather retanning. This retanning agent shows remarkable fluorescent stability compared to other leather chemicals, and the resultant leather exhibited an obvious fluorescence effect [86]. Zhang presented a facile and green approach to prepare phosphorus-nitrogen-containing waterborne polyurethane/graphene nanocomposite as flame-retardant retanning agent [87]. Longer flameless combustion time and higher limit oxygen index (LOI) values of leather retanned by flame-retardant WPU denote the enhanced flame retardancy and ameliorated smoke suppression.

Another interesting report recently emerged involved a polyurethane retanning agent with the function of reducing free formaldehyde in leather [88]. Chromotropic Acid (CA) from was used as a chain extender to synthesize this retanning agent (CAGAPU). Free formaldehyde can react with CA monomers, forming interaction of two naphthalene rings. As a result, CAGAPU has a positive effect on reduction of free formaldehyde during the retanning process in leather manufacture. (Figure 13) The work provides an efficient way to solve the trouble of free formaldehyde in leather.

### 3.2. Finishing

Finishing is able to modify the shade/gloss/handle, hide any defects or irregular appearance, improve physical properties, and offer functional properties to the final leather. A series of mechanical operations are carried out during this processes. Normally, polymeric coatings are applied to the leather surface. Among diverse alternatives, PU has recently become the most widely used coating-forming material. For instance, Wang reported an anti-biofouling PU with zwitterionic sulfobetaine side groups in leather coatings [89]. The incorporation of zwitterionic groups into the hard segment of polyurethane created more hydrogen bonding and polar interactions within this region, and thus made the hard components more thermodynamically and thus incompatible with the soft segments. The presence of zwitterionic group results in good anti-mold adhesion performance of polyurethane coatings. The appearance of this coating may extend the lifespan of the leather product. Although it has antimicrobial adhesion effect, these zwitterionic polyurethanes are not bactericidal. In this regard, Xu [90] covalently conjugated sulfanilamide (SA) into PU backbone as chain extender, and yielded a PU with enzymatically switchable antimicrobial capability as leather-finishing material. The contaminant-derived urease was utilized as trigger targets, SA can be released from the covalent bonded PU coatings. Without urease, the SA-conjugated PU coating displayed excellent hydrolytic resistance, which stopped any SA release from the conjugation. The antimicrobial ability was able to rapidly switch on, in case of exposure to urease. This freely delivered SA by urease-catalyzed break of urea linkages (Figure 14).

We prepared a diverse color-tunable luminous polyurethane leather coating (CLPU) which can reversibly change color as well as fluorescent emission through the UV–Vis or UV–darkness circle, via covalent incorporation of amino-functionalized phosphors and photochromic 1-(2-hydroxyethyl)-3,3-dimethylindolino-6′-nitrobenzopyrylospiran (SP) [91]. Before UV irradiation, SP was colorless, which allowed the phosphors green light to pass through the polyurethane matrix. The colored open-ring merocyanine (MC) induced by UV irradiation formed an optical filter and shielded most of the energy from the phosphors. As a result, only part of the luminescent can pass through resulting in an orange coloration. Finally, the finished leather exhibited diverse color-tunable luminous phenomena. In a different study, a photosensitive silicone-containing polyurethane acrylate resin (Si-IPDI-HEA) was prepared for leather finishing [86]. The properties of polysiloxane and polyurethane are maintained. Besides, polymerization of Si-IPDI-HEA can rapidly realized under UV irradiation. The import of silicon into pre-polymer could improve the thermostability. The dispersion surface energy of the UV-cured film can be reduced by the change of microstructure as well.

## 4. Summary and Outlook

Since first prepared by Bayer in the 1940s, polyurethane has been one of the most common, versatile, and researched materials in the world. PUs can be produced from a variety of diisocyanates, polyols, chain extenders, and crosslinking agents, making it possible to obtain a wide range of tailored materials. There is no doubt that functional PU, such as anti-fouling PU, self-healing PU, antibacterial PU, luminescent and color-tunable PU, as well as shape memory PU, will be the main research points in both academic and commercial fields. Even if a lot of progress has made during the last few decades, various new and intriguing challenges remains to be resolved. For example, the demand for PU products is increasing day by day, so recyclability of the product is of great significance. Majority of studies on the use of vegetable oils as substitutes to petroleum based materials for PU production. Nevertheless, there are certain drawbacks associated with these kinds of materials especially in regard to performance. Recent advances have focused on bi-functional PU, even multifunctional PU. The enormous number of available functional PU open unlimited possibilities for the development of advanced leather products. We expect that these concepts may steer innovative, smart, intelligent, environmental, and recycling materials to be used in daily life.

## Figures and Tables

**Figure 1 polymers-12-01996-f001:**
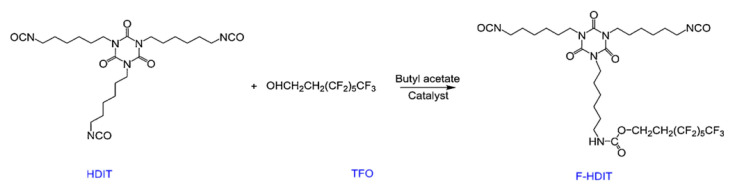
Structures of fluorine unit for preparing polyurethane (PU) was endowed with low surface free energy and great wetting ability [15].

**Figure 2 polymers-12-01996-f002:**
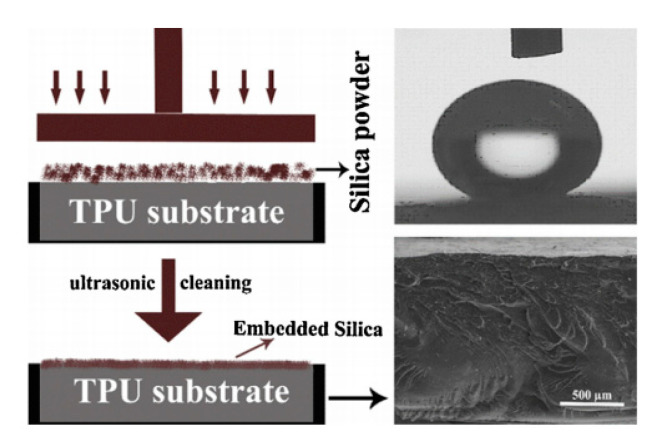
Schematic representation of the pressing method for fabrication of thermoplastic polyurethane/silica (TPU/silica) coatings (**left**) and cross-sectional morphology along with the water drop profile for the superhydrophobic sample (**right**) [18].

**Figure 3 polymers-12-01996-f003:**
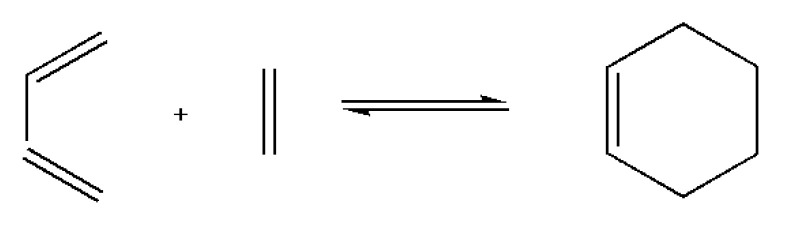
General equation of the Diels–Alder reaction.

**Figure 4 polymers-12-01996-f004:**
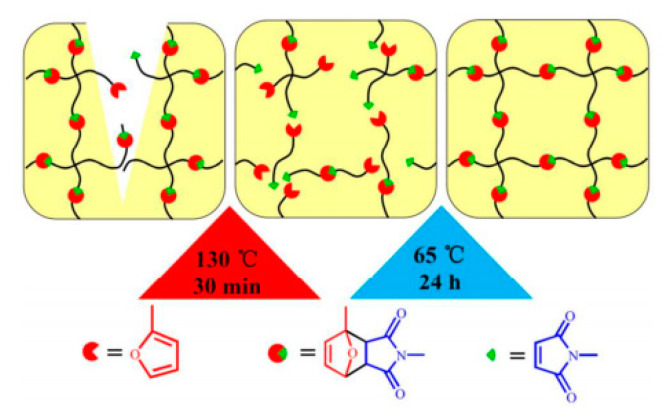
Schematic illustration of the self-healing mechanism of WPU-DA-x [36].

**Figure 5 polymers-12-01996-f005:**
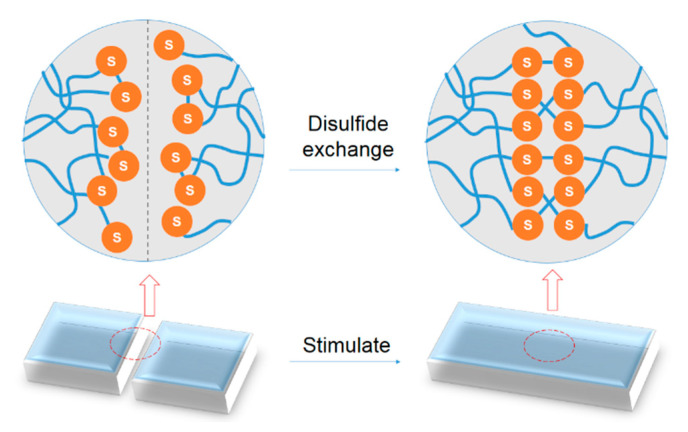
Healing illustration of self-healing polyurethane based on the combination of disulfide bonds and shape memory effect.

**Figure 6 polymers-12-01996-f006:**
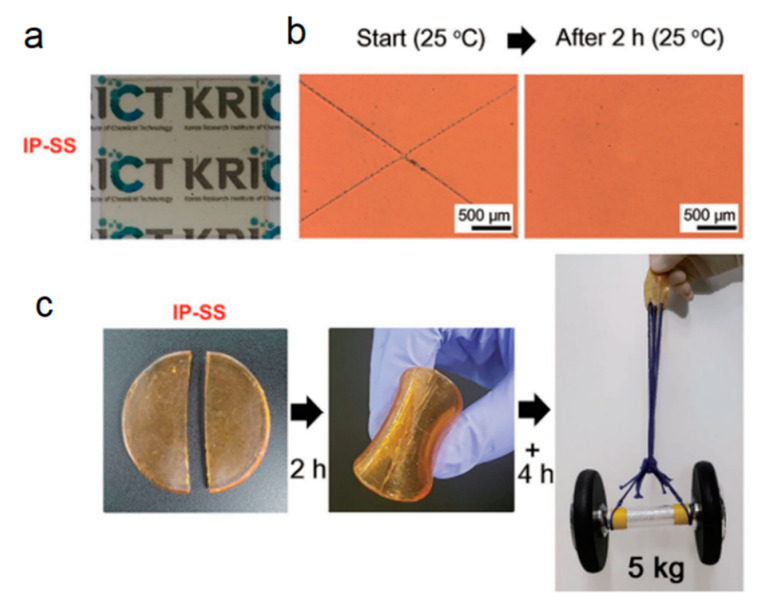
(**a**) Photograph of the TPU film (25 mm × 25 mm × 0.3 mm) of IP–SS. (**b**) Optical microscopy images of the X-shaped scratch on the TPU film of IP–SS before and after healing for 2 h at 25 °C. (**c**) IP–SS film cut in half, respliced, and healed for 2 h (+4 h) at 25 °C, followed by a 5 kg dumbbell lifting test [45].

**Figure 7 polymers-12-01996-f007:**
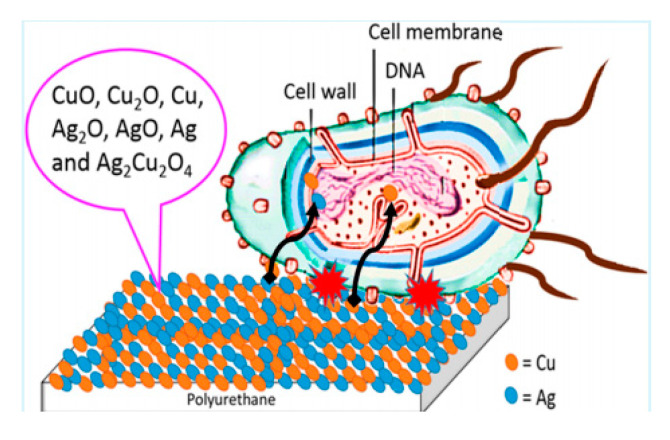
Antibacterial illustration of Cu–Ag-sputtered polyurethane (PU) catheters [54].

**Figure 8 polymers-12-01996-f008:**
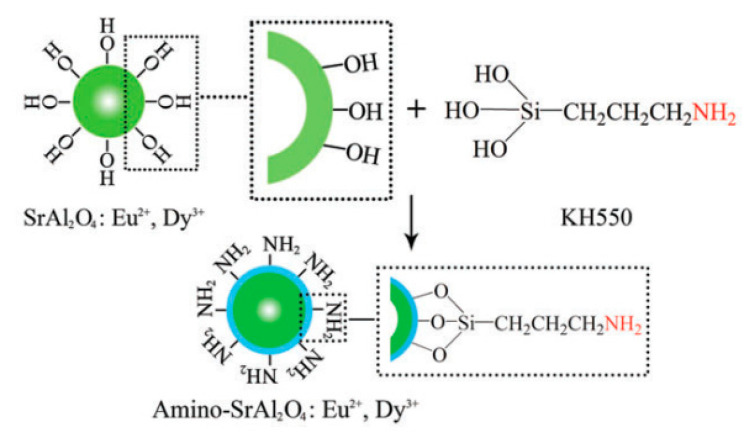
Synthesis route and molecular structure of Amino-SrAl_2_O_4_:Eu^2+^,Dy^3+^ [66].

**Figure 9 polymers-12-01996-f009:**
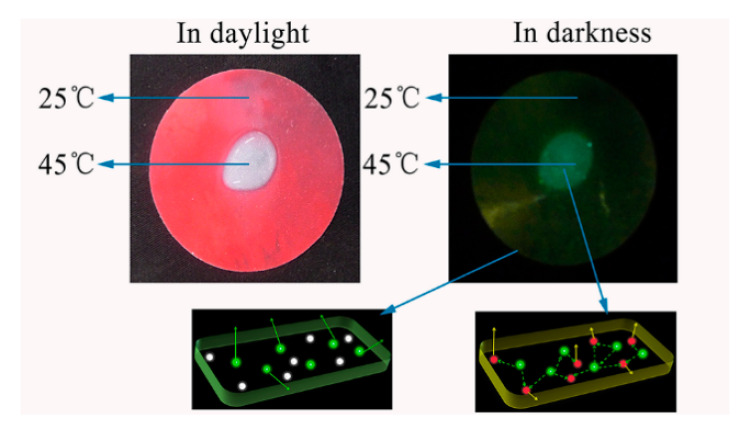
The digital image of thermochromic luminous phenomena and thermosensitive process together with luminescence model in darkness [70].

**Figure 10 polymers-12-01996-f010:**
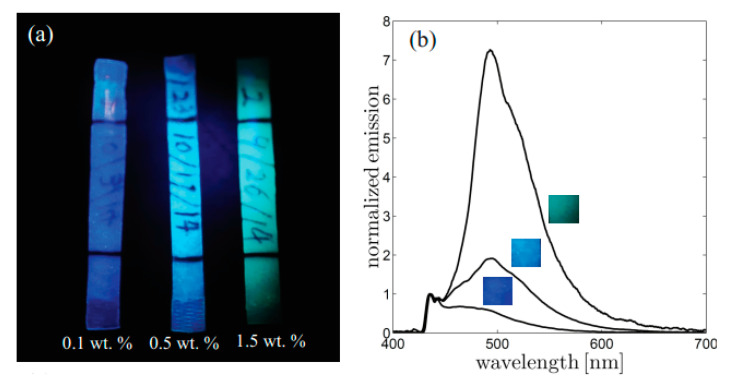
(**a**) Photographic image of three TPU-BBS samples with concentrations 0.1, 0.5, and 1.5 wt.%. The image is obtained through a commercial linear plastic polarizer. (**b**) Fluorescence spectra for the three concentrations are reported in the graph. Spectra are normalized so that the emission is equal in correspondence of the monomer peak at 436 nm. The fluorescence emission of dye aggregates increases with dye concentration and is associated with the progressive shift of material coloration from blue to green [73].

**Figure 11 polymers-12-01996-f011:**
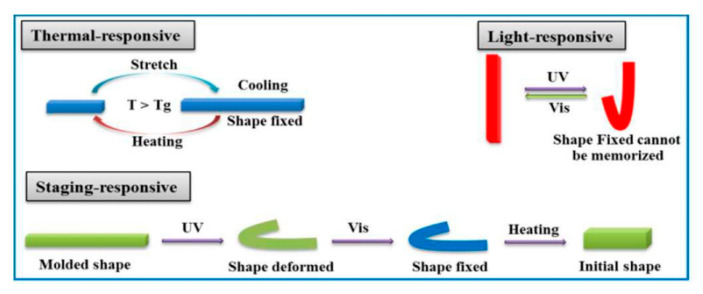
Illustration of the staging-responsive shape memory effect (SME) compared to the thermally responsive and light-responsive SMEs. (A traditional TSMP is deformed into a temporary shape and then recovered by “hot programming”, while a traditional LSMP can be deformed under UV light and promptly recovered using another light stimulus, e.g., Vis light. For the SR-SMPs, samples were first directionally oriented. They spontaneously deformed into a temporary shape by “UV programming”, and this deformed shape was unchangeable even under irradiation by Vis light. Only “hot programming” allowed the initial shape to be recovered [82].)

**Figure 12 polymers-12-01996-f012:**
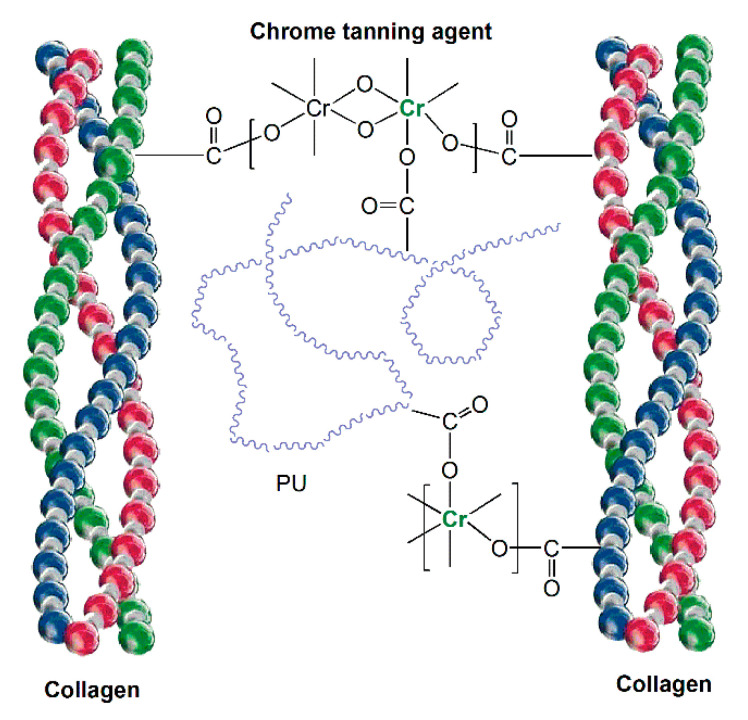
Mechanism of polyurethane retanning agent.

**Figure 13 polymers-12-01996-f013:**
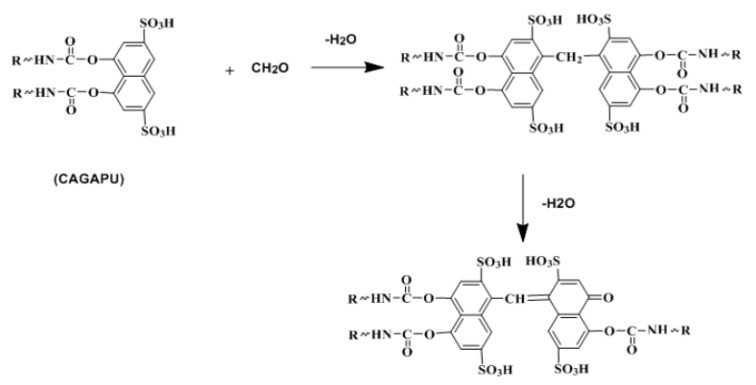
Reaction mechanism of CAGAPU with formaldehyde [88].

**Figure 14 polymers-12-01996-f014:**
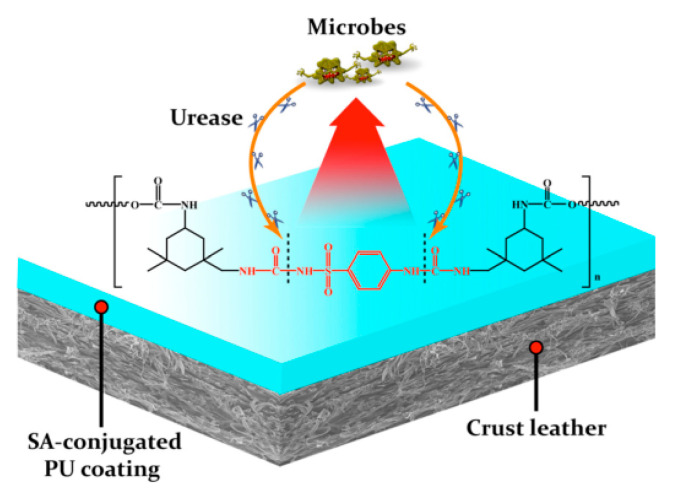
Schematic illustration of possible mechanism by which the SA-conjugated PU leather coating functions [90].

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
