# Peer review of "Recent Advances in Functional Polyurethane and Its Application in Leather Manufacture: A Review"

_polymers, 2020, doi:10.3390/polym12091996_

Round 1

Reviewer 1 Report

The manuscript makes a detailed review of the application of polyurethanes in leather manufacturing. The manuscript does not provide new knowledge. The evaluator thinks that if a systematic review of the subject is carried out, more valuable conclusions could be generated with respect to the subject discussed (which in the opinion of the evaluator is quite relevant).

Author Response

Dear reviewer,

Thanks for your comments concerning our manuscript entitled Recent advances in functional polyurethane and its application in leather manufacture: a review”. Your comments are valuable and helpful for improving my paper. Revisions are marked in red. The respond to your comments is as following:

 Point

The manuscript makes a detailed review of the application of polyurethanes in leather manufacturing. The manuscript does not provide new knowledge. The evaluator thinks that if a systematic review of the subject is carried out, more valuable conclusions could be generated with respect to the subject discussed (which in the opinion of the evaluator is quite relevant).

Response: Thank you for your encouragement to my work. I tried my best to improve the manuscript and made some changes in the manuscript.  These changes will not influence the content and framework of the paper.

I really appreciate for reviewers’ warm work earnestly, and hope that the correction will meet with approval.

Once again, thank you very much for your comments and suggestions.

Reviewer 2 Report

Dear Editor, in this review all Recent advances in functional polyurethane and its application in leather manufacture have been described. The review is well organized and provides some new and interesting properties and applications of PU. For this reason I propose to accept it for publication. In following you can find some of my comments.

Anti-fouling properties of polymers are mainly due to existence of zwitterionic groups. However, such works in PUs have been not mentioned in the first part of this review. On the contrary some works, which are not related with anti-fouling properties have been extensively described. This should be corrected in the revised manuscript.

Author Response

Dear reviewer,

Thanks for your comments concerning our manuscript entitled Recent advances in functional polyurethane and its application in leather manufacture: a review”. Your comments are valuable and helpful for improving my paper. Revisions are marked in red. The respond to your comments is as following:

Point

Dear Editor, in this review all recent advances in functional polyurethane and its application in leather manufacture have been described. The review is well organized and provides some new and interesting properties and applications of PU. For this reason I propose to accept it for publication. In following you can find some of my comments. Anti-fouling properties of polymers are mainly due to existence of zwitterionic groups. However, such works in PUs have been not mentioned in the first part of this review. On the contrary some works, which are not related with anti-fouling properties have been extensively described. This should be corrected in the revised manuscript.

Response: Thank you for your encouragement to my work. Plenty of works have been devoted to the anti-fouling polymers by introduce of zwitterionic groups. However, no articles about anti-fouling polyurethane with zwitterionic groups were reported in recent 3 years. So it is not mentioned in my manuscript. My thanks to for the reviewer’s recommendation. I tried my best to improve the manuscript and made some changes in the manuscript. These changes will not influence the content and framework of the paper.

I really appreciate for reviewer’s warm work earnestly, and hope that the correction will meet with approval.

Once again, thank you very much for your comments and suggestions.

Round 2

Reviewer 1 Report

the authors made the suggested changes

Reviewer 2 Report

Dear editor,

In the revised manuscript the proposed comments have been taken into consideration. For this reason I propose to accept it for publication.